# Sparse Progress and Risk Features in OpenVLA: Fragile Under Task-Family Shift

Joy Zheyun Yang[*]
Department of Computer Science
University of Oxford
Oxford, United Kingdom
Email: joy@robots.ox.ac.uk

Socrates Osorio[*]
Electrical Engineering and Computer Sciences
University of California, Berkeley
Berkeley, CA, USA
Email: socratesj.osorio@berkeley.edu

*Abstract*—RL fine-tuning of vision-language-action (VLA) policies depends on internal state that a scalar reward never names: whether a partial rollout is making progress, whether recovery is still possible, and whether optimization is drifting toward an unsafe shortcut. Rather than propose a new RL algorithm, this paper *measures* what one VLA policy, OPENVLA, represents internally during closed-loop LIBERO manipulation, and where that representation breaks. We cache residual-stream activations, train BatchTopK sparse autoencoders (SAEs), and audit the feature basis. The headline result is a failure mode under task-family shift. In distribution the decomposition is clean: a cross-validated progress readout reaches $0.932$ AUROC ($+0.221$ over the strongest motion-telemetry shortcut, decircularized top-20: $0.888$), and progress and risk occupy nearly disjoint feature sets (Jaccard $0.020$). But under *family-held* splits (train `goal`+`object`, test `spatial`+`long`) the same readout collapses to $0.537$ AUROC, near chance, while leave-one-task-out *within* families stays at $0.933$ (Appendix P reconciles every split). The central takeaway is this $0.933 \rightarrow 0.537$ family-level collapse: internal-feature reward/value signals survive task-level shift but not family-level shift, so any such signal must be re-validated on the target task *family* before RL use. Within scope (OPENVLA/LIBERO, all hazard labels are simulator-derived telemetry proxies, *not* real-world safety measures) the features power bounded RL-facing signals: a potential-based dense-reward pilot ($0.882$ AUROC return recovery from a $25\%$ prefix vs. $0.868$ motion-only), feature-stratified offline curation ($+0.161$ AUROC over return-only filtering, $21\times$ more recovery data), and a live monitor veto cutting violations $16\%$ at $92\%$ of baseline success, or up to $27\%$ at $86\%$, while feature clamping fails a specificity test against matched random controls, which we report as a negative result. New in this version, the loop is closed on weights, and yields a negative result we foreground: in a sparse-return offline fine-tuning run (LoRA, potential-weighted self-imitation), weighting steps by an SAE-potential advantage lets the update consume *all* rollouts, including the failures return-only filtering discards, yet closed-loop strict success matches rather than beats the sparse-return baseline ($0.650$ vs. $0.680$, base $0.670$, $N{=}200$/arm), even though the same readout is near-ceiling as a diagnostic ($0.985$ AUROC). Decodability of progress does not yet imply trainability on it. The contribution is a measurement and a recipe: report the auditable internal representation of progress and risk, and re-validate it under distribution shift, which RL fine-tuning *can* create when the training and target task families differ, though we do not directly measure the shift an RL run induces.

## I. INTRODUCTION

Large vision-language-action (VLA) policies have made imitation-based robot learning more general: a single model can condition on language, images, and histories to produce actions across many tasks [6, 31, 17, 4]. Yet imitation learning alone leaves important capabilities underspecified. Demonstrations overweight successful behavior, underrepresent recovery, and optimize action likelihood rather than downstream objectives such as task completion, speed, resource use, human preference, or safety. This motivates RL fine-tuning for VLA models: learn from failures, optimize task-level rewards, and adapt to deployment distributions that differ from curated demonstrations.

RL for VLAs poses distinctive difficulties. The reward is often sparse and semantic, credit assignment passes through perception, language grounding, and continuous control, and real-world exploration is constrained by resets, hardware wear, and human safety. A scalar reward reports whether an episode succeeded, but not whether the policy was internally tracking progress, whether it had entered a recoverable state, or whether an unsafe shortcut was being reinforced. This is especially problematic for offline or batch RL, where all improvements depend on the information already present in logged trajectories. If the policy representation does not expose progress and risk, then value functions and reward models built on it can learn brittle correlations rather than meaningful control state.

This paper studies a representation-level prerequisite for RL-ready VLAs: *can we identify sparse internal features that encode progress and risk during closed-loop execution?* We do not present a new RL algorithm, and we do not claim that sparse features alone solve safe fine-tuning. Instead, we argue that RL-oriented VLA pipelines should include an internal audit before fine-tuning. Such an audit asks whether a policy's hidden states contain decodable concepts for partial progress, likely future violation, and intervention-relevant directions. These quantities are useful for reward shaping, offline dataset triage, conservative policy improvement, and hazard-proxy monitors that flag likely violations before they occur.

We instantiate this audit with SAFESAE-VLA. We run OPENVLA in LIBERO manipulation environments [20], cache residual-stream activations at layers 16, 20, and 24, train

**Sparse VLA Progress Analysis Pipeline**

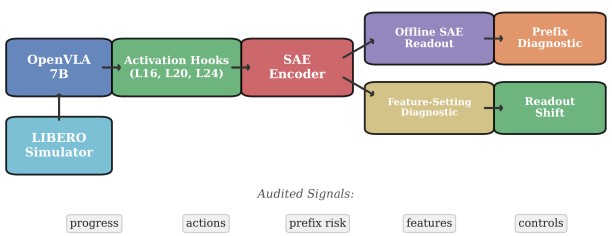

*Audited Signals:*

progress     actions     prefix risk     features     controls

Fig. 1: SAFESAE-VLA audits VLA policies before RL fine-tuning by caching internal activations, decomposing them into sparse features, and testing whether the features support progress readouts, prefix-risk monitoring, and feature-setting sanity checks.

BatchTopK SAEs [8], and evaluate sparse feature readouts. Our main positive result is that geometric task progress is strongly and compactly represented. A cross-validated linear readout over layer-20 SAE features reaches 0.932 AUROC (95% CI $[0.905, 0.954]$) for high- vs. low-progress episodes, and a top-20 sparse subset whose features are ranked *inside each training fold*, removing the circularity of evaluating on the same data used to select features, retains 0.888 AUROC. Critically, both readouts dominate the strongest motion-telemetry shortcut (end-effector velocity, 0.711) by more than 0.17 AUROC, so the progress signal is not merely a restatement of gross geometry. This is the kind of representation an offline RL or reward-learning pipeline would want: progress is not only present in raw hidden state, but concentrated in inspectable directions that can be ranked, stress-tested, monitored, and, as we show, perturbed.

    Crucially, the audit does not stop at decoding: the progress feature acts as a usable *dense reward*. As a potential-based shaping signal [25] read from the first quarter of a rollout, it lifts a motion-only value estimate from 0.868 to 0.882 AUROC and predicts an object-lift subgoal at 0.973 (Section V-E). Three policy-level signals follow: feature-stratified curation keeping 21× more recovery data (Section V-D). A monitor veto cutting the live violation rate 0.711→0.598 at 92% success (Section VI-B). And an offline RL run in which the same potential supplies step-level credit on a real LoRA weight update, without, we report plainly, a return gain over the sparse-return baseline (Section V-F).

    We are equally explicit about four honest negatives. (0) **Decodability is not trainability**: converting the near-ceiling progress readout into offline-RL step weights yields no return gain over a sparse-return baseline (Section V-F). (i) **Task-family shift is the paper's headline finding and the load-bearing negative:** under a *family-held* split (train `goal+object`, test `spatial+long`) the full progress readout collapses $0.932 \rightarrow 0.537$ AUROC, while surviving leave-one-task-out *within* families at 0.933 (Appendix P), so any reward/value use must be re-validated on the target task *family*. (ii) **Hazard-proxy detection is weak**: the sparse

readout loses to a raw MLP on AUROC (0.632 vs. 0.640) and is at chance for high-approach speed (0.535). We therefore do *not* claim strong detection. We use it only as a *rank-and-halt* control (Section VI-B), justified operationally by its low 0.009 false-alarm rate rather than by discrimination accuracy. (iii) **Feature clamping** shows behavioral sensitivity but *not* causal specificity, targeted features are not reliably better than matched random controls, so we retract any feature-level guardrail claim. Each negative names a concrete next experiment rather than hiding a weakness.

**Contributions.** Our single crisp contribution is a *measurement*: a statistically grounded characterization of what OPEN-VLA internally represents about progress and risk, and, uniquely for RL4VLA, *where that representation breaks under task-family shift, the kind of distribution shift RL fine-tuning can create*. Concretely: (1) **Headline:** OPENVLA's progress readout *collapses to chance under task-family shift*, 0.933 AUROC leave-one-task-out within families vs. 0.537 family-held, so an internal reward/value signal that looks strong in distribution can be near-useless one family away, and must be re-validated per family. (2) We frame sparse internal feature audits as a pre-RL evaluation layer for VLA policies, with statistically grounded evidence (bootstrap CIs, de-circularized readouts, telemetry baselines) that OPENVLA concentrates progress in a compact sparse basis nearly disjoint from its risk basis (Jaccard 0.020), a diagnostic a black-box readout would hide. (3) We provide *four* RL signals that consume the features: (a) a potential-based dense-reward pilot ($0.868 \rightarrow 0.882$ AUROC at a 25% prefix, object-lift subgoal 0.973, return-invariant under the terminal-potential condition $\Phi(s_{\text{terminal}})=0$ [25]). (b) feature-stratified offline curation (+0.161 AUROC over return-only filtering, 21× more recovery data). (c) a live rank-and-halt monitor veto cutting simulated proxy violations 0.711→0.598 at 92% of baseline success. And (d) a sparse-return potential-weighted self-imitation run on weights in which the SAE potential supplies step-level credit, reported as a *negative*: no gain over the sparse-return baseline (Section V-F). (4) We stress-test a prefix *hazard-proxy* monitor (simulator telemetry, not real safety) and report it operationally, false-alarm budget and per-category coverage, since a raw MLP wins on aggregate AUROC. (5) We report closed-loop clamping as a *negative* specificity result, matched random controls move behavior comparably, naming closed-loop specificity as the central open problem rather than claiming a guardrail.

## II. RELATED WORK

**RL fine-tuning for VLA policies.** A growing line of work moves VLA models beyond imitation into RL. Online RL improves OPENVLA-style policies on manipulation tasks [15, 21], consistency-policy and offline-to-online recipes stabilize fine-tuning [9], and RL has been used to turn vision-language models into decision-making agents [30]. These methods optimize scalar task or preference reward. None inspect whether the policy's internal state already encodes the progress and risk concepts the reward is meant to capture. We are com-

plementary: an audit layer that runs *before* such fine-tuning and provides dense internal state for shaping, curation, and monitoring.

**Reward hacking and learned-reward failure.** RL is only as good as its reward. Reward misspecification produces qualitatively wrong behavior [26], reward models over-optimize under scaling pressure [12], and reward hacking has been formally characterized [27]. A recurring theme is that a high-return policy can exploit a proxy that diverges from the intended objective. Because our progress proxy is geometric, it is itself hackable. We therefore treat sparse features as a way to *audit* a learned reward (does it ignore internal progress? does it reward motion shortcuts?) rather than as a reward to be maximized directly.

**Sparse autoencoders and probing.** Linear probes recover concepts from intermediate activations [1], and sequence models learn emergent, often linear, world representations [19, 24]. SAEs decompose superposed activations into sparser, more monosemantic features [5, 10, 28, 13], with BatchTopK improving the sparsity/fidelity trade-off [8]. Most SAE work targets language models. Interpretability and steering of *robot policies* with sparse internal features is far less explored [16, 22]. Mechanistic interpretability has been argued as a pillar of AI safety [3]. We bring this toolkit to closed-loop VLA execution and ask, specifically, whether the recovered features are useful for RL rather than for explanation alone.

**Safe robot RL.** Safe RL adds shields, recovery policies, and constraints around the learner [14, 11, 29, 7]. These operate at the action or constraint level. We study a complementary internal layer: whether the policy's own hidden state warns of future violation early enough to filter, stop, or steer, and whether perturbing it changes safety outcomes.

## III. WHY RL FOR VLAs NEEDS INTERNAL AUDITS

Throughout, all hazard and "safety" quantities are *simulator-derived telemetry proxies*, not real-world safety events (detailed in Section VI). No result here is a deployment safety claim. Safety and sample efficiency are coupled in robot RL [14, 11, 7], and VLA policies intensify the coupling: one token-conditioned policy executes many semantically different tasks, so a collision near a target can be useful contact or catastrophic depending on instruction and phase. A scalar reward compresses these distinctions after the fact, whereas RL updates need credit at intermediate states. This is exactly where internal features help: they can reveal whether the policy separates moving toward a goal from merely moving fast.

This matters most offline, where VLA fine-tuning typically begins [18]. That pipeline is vulnerable to three failure modes: success-heavy data with little recovery, negatives too sparse to fit robust value functions, and rewards that attach to motion rather than to progress. Sparse feature audits make these risks measurable: if a few features decode progress, an offline system gains a state variable to shape rewards or stratify data. If the readout collapses under family-held splits (Section V), a reward built on it is likely learning shortcuts. Table XII (Appendix B) summarizes this interface, sparse features as

auditable state variables, not a black-box replacement for rewards.

## IV. SPARSE FEATURE AUDIT

### A. Rollout Collection and Activation Caching

We evaluate OPENVLA on LIBERO manipulation rollouts spanning `goal`, `object`, `long`, and `spatial` task suites. During closed-loop execution, we cache residual-stream activations at transformer layers $\ell \in \{16, 20, 24\}$. For OPENVLA, each cached tensor has shape $[T, 7, 4096]$, where $T$ is the control horizon and the token dimension corresponds to the model's action-relevant representation. Our primary analysis uses layer 20 because it is a middle-to-late representation with strong progress readout performance and continuity with the broader SAFESAE-VLA pipeline.

Let $h_t^{(\ell)} \in \mathbb{R}^{4096}$ denote an aggregated hidden state at timestep $t$ and layer $\ell$. A BatchTopK SAE encodes $h$ into sparse feature activations $z \in \mathbb{R}^{d_{\text{sae}}}$ and reconstructs $\hat{h}$:

$$z = \text{TopK}_{\text{batch}} \left( \text{ReLU}(W_{\text{enc}} h + b_{\text{enc}}) \right), \quad \hat{h} = W_{\text{dec}} z + b_{\text{dec}}. \tag{1}$$

The primary dictionary uses $d_{\text{sae}} = 16{,}384$ and $k = 32$. We also maintain a 32K layer-20 ablation and layer-16/layer-24 dictionaries for health checks, but the main results focus on the 16K layer-20 model.

### B. Progress Target for RL-Relevant State

An RL-ready diagnostic should expose partial completion, not only terminal success. We therefore construct a high-contrast progress target from simulator telemetry. For each episode, we compute a suite-normalized geometric progress score based on task-specific end-effector and object state. The top quartile within each suite is labeled high-progress, the bottom quartile is labeled low-progress, and middle quartiles are excluded from the main binary readout. This yields a balanced but deliberately high-contrast audit set: it asks whether the representation distinguishes clearly useful progress from clearly poor progress.

This target is not a perfect reward. It is geometric rather than semantic, and it is not meant to replace human preference or task completion. Its value is that it provides a dense state variable that a reward-shaping or offline RL pipeline might use as an auxiliary signal. If sparse features cannot recover this variable, then expecting them to support more nuanced RL objectives would be premature.

### C. Prefix Safety Target

We evaluate safety on a telemetry-audited benchmark with five fixed hazard categories: collision, excessive force, boundary violation, high approach speed, and object drop. At each prefix, the monitor observes only information up to the current timestep and predicts whether a violation occurs within a future window of 25 control steps. Evaluation uses leave-one-task-out splits, so performance reflects task-held generalization rather than memorization of a trajectory family.

This target matches an RL safety use case: before updating or deploying a policy, we want to know whether its internal

state warns of future failure early enough to filter, stop, or recover. Lead time and false alarms are therefore as important as AUROC. A high false-alarm monitor may make fine-tuning too conservative, while a category-blind monitor may allow reward hacking in an unmonitored failure mode.

### D. Feature Ranking and Readouts

We aggregate sparse activations at the episode level and test each active feature for differential activation between high- and low-progress episodes using Mann–Whitney tests [23]. We control false discovery with Benjamini–Hochberg correction [2] and rank features by a score combining effect magnitude and adjusted significance. We then evaluate logistic readouts on the full SAE feature vector and on top-ranked sparse subsets.

The ranking is used in two ways. For prediction, we avoid overclaiming by reporting held-out or task-held readouts. For inspection and perturbation, we use globally ranked top features as candidate internal concepts. This distinction is important: a feature list selected on all data is useful for hypothesis generation and sanity checks, but not an unbiased estimate of monitor performance.

## V. PROGRESS FEATURES FOR RL STATE

**Primary protocol (one box, all denominators).** Model: OPENVLA-7B LIBERO-finetuned checkpoints (one per suite), frozen unless stated. Sim: LIBERO `goal`/`object`/`long`/`spatial`, official protocol. Data: 750 closed-loop rollouts. The *headline audit set* is the within-suite high/low binary split ($n$=374 episodes, per-suite counts in Table XXI and *all* episode counts consolidated in Table XXII, Appendix L). Features: layer-20 BatchTopK SAE (16,384, $k$=32). *Headline numbers*: 0.932 episode-CV AUROC (powered suites `goal`/`object`) and the 0.537 family-held collapse (Table III). Everything else, continuous target (750 eps), safety/veto (560, leave-one-task-out), curation, shaping, clamping, is supporting evidence at its stated $N$. The offline RL loop (Section V-F) uses a separate 300-rollout Object set, 100 held-out-init eval episodes per arm.

### A. Progress Is Compactly Decodable

Table I gives the central representation result with three additions the prior version lacked: bootstrap confidence intervals, motion-telemetry shortcut baselines *in the same table*, and a de-circularized readout. A cross-validated logistic readout over the 16K SAE features reaches 0.932 AUROC (95% CI [0.905, 0.954], the in-sample point estimate is 0.918). The key shortcut control is motion telemetry: because the progress label is geometric, end-effector velocity alone is predictive, but it reaches only 0.711 AUROC and action magnitude only 0.572. The SAE readout beats the strongest telemetry baseline by +0.221 AUROC, so the sparse features encode progress information beyond gross motion, exactly the property a reward model needs to avoid rewarding motion shortcuts.

TABLE I: Progress readout on 750 OPENVLA rollouts (within-suite high/low split, $n$=374 episodes). AUROC CIs are 2000-sample bootstrap. "held-out-ranked" selects the top-20 features inside each training fold to avoid circular evaluation. Motion-telemetry rows are the shortcut baselines.

| Method | AUROC (95% CI) | F1 | Prec. | Recall | PR-AUC |
|---|---|---|---|---|---|
| SAE Feature LR (16K) | **0.932** [.905,.954] | **0.817** | 0.797 | **0.839** | **0.913** |
| Top-20 (in-sample) | 0.894 [.860,.924] | 0.752 | **0.844** | 0.679 | 0.885 |
| Top-20 (held-out-rank) | 0.888 [.857,.919] | 0.745 | 0.831 | 0.674 | 0.872 |
| EE velocity (telemetry) | 0.711 [.659,.761] | 0.661 | 0.652 | 0.671 | 0.690 |
| Action magnitude | 0.572 [.514,.629] | 0.560 | 0.571 | 0.549 | 0.566 |
| Random features | 0.554 [.495,.612] | 0.550 | 0.566 | 0.536 | 0.563 |

Two readout variants address the circularity concern that features ranked on all data and then evaluated on the same data inflate performance. The *held-out-ranked* top-20 readout ranks features by Mann–Whitney effect *inside each training fold only* and never sees the test split during selection. It retains 0.888 AUROC ([0.857, 0.919]). The full-vs-compact gap is small but significant: a paired bootstrap gives $\Delta$=0.044 ([0.027, 0.061]), positive in 100% of resamples. So the full dictionary is reliably, if modestly, better than 20 features, and a small inspectable set already captures most of the signal without circular feature selection.

**The** 0.932 **is not just the easy high-vs-low split.** A fair objection is that the headline drops the middle progress quartiles, deliberately creating an easy high-contrast target. We therefore also fit a *continuous* progress regression on *all* 750 episodes with no quartile dropping (Table XVII). The full SAE attains Spearman 0.739 ($R^2$=0.596) under episode CV and 0.676 ($R^2$=0.428) under task-held (leave-task-out) splits, a real but more modest signal that mirrors the binary number's behavior (strong within distribution, degraded as shift grows), while rich motion+suite telemetry dominates the continuous target outright (Spearman 0.98). The continuous result thus tempers, rather than inflates, the claim: SAE features carry genuine graded progress information, but the 0.932 should be read as the separability of clear progress from clear failure, not as semantic decoding of every intermediate state.

The claim is representational, not that SAE features are the only progress predictor: OPENVLA's internal state contains a sparse, inspectable basis aligned with partial progress (Figure 2) that beats the telemetry shortcut, is cheap to log on every fine-tuning checkpoint, and can audit a learned reward (a reward that improves return while these features stay flat is likely exploiting a shortcut). Per-suite Hanley–McNeil CIs (Table II) show only `goal` and `object` are adequately powered (CIs $\leq \pm 0.04$). `long`/`spatial` CIs span $\pm 0.19$–$0.31$, so **we restrict every headline progress claim to the two powered suites**. A leave-suite-out check confirms the aggregate is theirs (removing `goal`$\to$ 0.905, `object`$\to$ 0.918, removing either small suite leaves it $\approx$ 0.93).

### B. Differential Features Are Active and Structured

Differential analysis identifies many progress-associated dimensions rather than a single brittle neuron. In the main layer-20 audit, a large fraction of active SAE features show

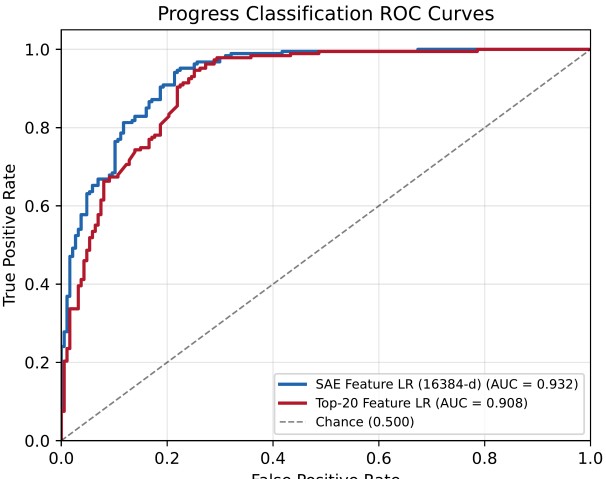

Fig. 2: ROC curves for progress readouts on the within-suite high/low split. The full 16K SAE readout and the compact top-20 sparse readout both dominate matched random-feature controls across the operating range, indicating that the progress signal is concentrated in a small, inspectable set of directions rather than smeared across the full dictionary.

TABLE II: Per-suite progress separation with Hanley–McNeil 95% AUROC CIs. Only the two data-rich suites (`goal`, `object`) are adequately powered. `long` and `spatial` CIs are too wide to interpret, so headline claims are restricted to the top two rows.

| Suite | AUROC (95% CI) | High/Low | Powered? |
|---|---|---|---|
| `goal` | **0.985** [.960,1.00] | 45/129 | yes |
| `object` | 0.947 [.910,.984] | 125/17 | yes |
| `long` | 0.700 [.515,.885] | 12/32 | no |
| `spatial` | 0.667 [.353,.981] | 5/9 | no |

significant differences after FDR correction. The strongest top features are not dead artifacts: high-progress features activate in roughly 57–72% of high-progress episodes and only 7–21% of low-progress episodes, while low-progress features show the mirrored pattern (full top-20 prevalence table in Appendix O, Table XXV).

For RL, the activity structure matters: features active in most high- but few low-progress episodes support trajectory relabeling and reward-model debugging, while low-progress features mark recovery states an offline pipeline should preserve rather than discard, selecting not only high-return trajectories but also informative low-progress ones.

### C. The Headline RL Finding: Progress Features Collapse Under Task-Family Shift

The most RL-relevant measurement in this paper is not the 0.932 but what happens to it under distribution shift. We take as a *working assumption* (not a measured claim) that RL fine-tuning often moves the policy toward task families under-represented at collection time. This need not hold universally: a replay buffer covering all target families would not induce family shift, and RL can also drift the policy's own *behavior* in ways our family-held splits do not measure. We test neither

TABLE III: *Family-held* (cross-suite) transfer for progress readouts: entire task families are held out, the hardest split we test. Sparse top features can transfer better than the full dictionary in one direction, but family shift remains far harder than random splits, and than leave-one-task-out *within* families (0.933, Appendix P).

| Split | Method | AUROC | Test N |
|---|---|---|---|
| `goal+object`→`spatial+long` | SAE LR | 0.537 | 58 |
| `goal+object`→`spatial+long` | Top-20 LR | **0.658** | 58 |
| `spatial+long`→`goal+object` | SAE LR | 0.586 | 316 |
| `spatial+long`→`goal+object` | Top-20 LR | **0.598** | 316 |
| Random split | SAE LR | 0.932±0.010 | – |
| Random split | Top-20 LR | 0.912±0.018 | – |

axis directly here and flag both as open. The collapse is specifically a *family-level* phenomenon, so the split granularity must be named precisely. Under *family-held* splits (train `goal+object`, test the held-out `spatial+long` family) the full SAE readout collapses from 0.932 to 0.537 AUROC, near chance. The top-20 subset transfers somewhat better (0.658), and the reverse direction is weak for both. Under *task-held* splits (leave-one-task-out, all families represented in training) the same readout holds at 0.933 (Appendix P reconciles all split granularities in one table). So related tasks in the training family carry the signal across task boundaries, but nothing carries it across families. We present this as a *finding*, not an apology: a policy fine-tuned toward a new task family can keep improving its scalar return while the features a reward shaper reads from drift to chance, baking in the wrong inductive bias undetected. The sparse, inspectable basis is what makes this diagnosable, re-ranking features *inside* the target family recovers signal, so the recipe is concrete: re-validate (and if needed re-rank) progress features whenever the target task *family* moves away from the collection distribution, a condition RL fine-tuning can create, but only when the training data does not already cover the target families.

### D. A Minimal RL Signal: Feature-Stratified Offline Curation

Every result so far is diagnostic. To give one concrete demonstration that the audited features *improve* an RL pipeline rather than merely describe it, we run a paired offline data-curation ablation, the simplest RL-relevant use, needing no online exploration. Mirroring offline RL, from a pool of logged OPENVLA rollouts we keep a fixed *budget* (50%) of trajectories to train a downstream progress readout, then test on a disjoint held-out set (30%). *Return-only filtering* keeps the highest-return episodes first (the success-heavy heuristic that discards low-return data). *feature-stratified curation* instead keeps a sample spanning the quantiles of the SAE progress-feature score, preserving recovery trajectories.

Over 20 splits (Table IV), return-only filtering retains almost no recovery data (2.4%) and, collapsing to a near single-class training set, yields only 0.761±0.175 AUROC. Feature-stratified curation retains 50.5% recovery data and reaches 0.921 ± 0.023 AUROC, a +0.161 gain, positive in 95% of splits. This is the missing link between "these features decode progress" and "these features help RL": ranking trajectories

TABLE IV: Offline data-curation ablation (20 splits, 50% keep budget). Feature-stratified curation preserves recovery data and trains a stronger downstream progress readout than return-only filtering.

| Curation strategy | Downstream AUROC | Recovery data kept |
|---|---|---|
| Return-only filtering | $0.761 \pm 0.175$ | 2.4% |
| Feature-stratified (SAE) | $\mathbf{0.921 \pm 0.023}$ | **50.5%** |
| $\Delta$ (strat. $-$ return) | **+0.161** (95% of splits) | +48.1 pts |

TABLE V: Reward-shaping / value-debugging pilot. The SAE progress potential, read from a short prefix, recovers the eventual return earlier than motion alone (sparse terminal reward, $\gamma$-discounted potential-based shaping). TPR is at a fixed false-positive budget. 72-ep. rows predict terminal success and the object-lift subgoal.

| Value signal (prefix) | Return AUROC | TPR@10% | TPR@20% |
|---|---|---|---|
| Motion only (25% prefix) | 0.868 | 0.702 | 0.814 |
| SAE progress $\Phi$ (25%) | 0.863 | 0.691 | 0.809 |
| Motion + SAE $\Phi$ (25%) | **0.882** | **0.729** | **0.851** |
| $\Phi \rightarrow$ success (72-ep.) | 0.834 | , | , |
| $\Phi \rightarrow$ object-lift (72-ep.) | **0.973** | , | , |

by an internal progress feature keeps the recovery behavior a return-only filter throws away.

### E. A Reward-Shaping Pilot: Progress as a Dense Value Signal

Curation improves a *data pipeline*. The headline RL premise is whether the progress feature can act as a *dense reward / value signal* that improves credit assignment when the true environment return is sparse. In LIBERO the env return is a single terminal success bit (Appendix H confirms `final_reward` $\in \{0, 1\}$ equals episode success), the worst case for offline RL [18]. We therefore run a value-debugging pilot in the potential-based-shaping framework of Ng et al. [25]: treat the calibrated SAE progress readout as a potential $\Phi$ that supplies a dense shaped reward $r' = r_{\mathrm{env}} + \gamma\Phi(s') - \Phi(s)$, and ask whether $\Phi$ lets a value/return estimate be recovered from a *short prefix*, i.e. earlier credit assignment, rather than only at the terminal step. Potential-based shaping is return-invariant by construction under the standard terminal-potential condition $\Phi(s_{\mathrm{terminal}})=0$, so the only question is whether $\Phi$ carries return information early.

Table V answers it on the held-out future-return benchmark. From only the first 25% of a rollout, the SAE progress potential recovers the eventual return at 0.863 AUROC alone and lifts a motion-only value estimate from 0.868 to 0.882 (85.1% vs. 81.4% of the outcome caught at a 20% false-positive budget), a real, if modest, dense-reward gain three-quarters of an episode before the sparse terminal reward arrives. On a separate 72-episode varied-success set, a progress value feature predicts terminal *success* at 0.834 AUROC and the *object-lift* subgoal at 0.973 (Appendix H). Scope: this is an offline value-prediction pilot, not an online run. We report the gain where the readout is calibrated and flag online shaping as the next step.

TABLE VI: Sparse-return offline RL on OPENVLA weights (LoRA), with vs. without the SAE-progress potential as a dense step-level advantage. Same data budget, steps, and optimizer. Strict success on held-out init states, pooled over two checkpoint generations. The $\Phi$ readout is near-ceiling as a diagnostic (episode AUROC 0.985) yet yields no return gain, decodability does not imply trainability.

| Arm (Object, 10 tasks) | Training data | Strict success | $\Delta$ vs base |
|---|---|---|---|
| Base policy (no update) | , | 67/100 | , |
| Sparse-return self-imitation | successes only (203 eps) | 136/200 | +0.010 |
| + SAE-potential advantage | all 275 eps, $\Phi$-weighted | 130/200 | −0.020 |

### F. Closing the Loop on Weights: SAE-Potential-Weighted Offline RL

The curation and shaping results above stop at data pipelines and value prediction. Here the features drive an actual *weight update*. We run sparse-return offline fine-tuning (potential-weighted self-imitation, a form of advantage-weighted behavioral cloning rather than a learned-value or Q objective) on a self-collected set of 300 OPENVLA rollouts on the full 10-task LIBERO-Object suite (30 inits/task, official protocol, terminal return = strict success only. 275 train / 25 validation episodes) and fine-tune the policy with LoRA on its own action tokens. The *sparse-return baseline* is success-filtered self-imitation: it can only use the 203 successful training episodes and discards every failure, exactly the recovery-destroying selection the curation ablation predicts is wasteful. The *SAE-potential arm* trains on *all* 275 training episodes, weighting each step by a potential-based advantage $\exp\{(\gamma\Phi_{t+1} - \Phi_t + r_t)/\tau\}$, where $\Phi$ is a calibrated SAE-progress readout fit on the training rollouts, the dense credit signal of Section V-E, now consumed by an optimizer instead of a prediction benchmark. Both arms share the LoRA budget, steps, and optimizer. Evaluation is strict closed-loop success on 10 held-out init states per task ($N$=100/arm).

Result (Table VI): **a negative result we foreground rather than bury**. Pooled over two checkpoint generations ($N$=200/arm, checkpoint variance $\sim\pm 0.1$), the $\Phi$-weighted arm reaches 0.650 vs. 0.680 for the sparse-return baseline (base 0.670), a small negative difference well within the checkpoint variance of $\sim\pm 0.1$, so we do not claim significance, even though the $\Phi$ readout itself is near-ceiling here (episode AUROC 0.985). The diagnostic is excellent, yet in this offline instantiation it does not convert into a return gain over simply filtering by sparse success. This separates two claims the literature conflates: *decodability* of progress (strong, Section V) and *trainability* on decoded progress (not demonstrated). Scope: one suite, offline, one weighting scheme. Online policy-gradient with the shaping term and the family-shifted replication remain open.

### VI. PREFIX HAZARD-PROXY MONITORING FOR RL FINE-TUNING

**Hazard labels are simulator-based proxies, not real-world safety measures.** Every hazard label in this paper is a *telemetry-derived proxy computed in simulation*, thresholded

TABLE VII: Prefix-only future-violation monitoring on the telemetry-audited benchmark. Lead is median lead time in control steps. False alarm is measured on successful episodes.

| Method | AUROC | PR-AUC | Lead | False Alarm |
|---|---|---|---|---|
| Raw Activation MLP | **0.640** | **0.535** | 51.7 | 0.079 |
| SAE Feature LR | 0.632 | 0.470 | 47.6 | **0.009** |
| Force Threshold | 0.615 | 0.477 | 40.0 | 0.023 |
| Telemetry LR | 0.587 | 0.487 | 49.5 | 0.043 |
| Random | 0.500 | 0.327 | 51.2 | 0.072 |

collision, force, boundary, speed, and drop signals, not a logged real-world safety event, and no number here should be read as a measure of real-world safety. We therefore call these *hazard-proxy* features rather than "safety features," and we caution that proxy AUROC need not transfer to real hazards. The monitoring and veto results below are evidence on a simulation proxy only. Real-hazard replication is required before any deployment safety claim.

### A. Future-Violation Monitoring

Safety-constrained RL requires warnings before violations, not post-hoc labels. Here "violation" means a simulated hazard-proxy event (see the caveat above). Table VII reports prefix-only future-violation prediction on the telemetry-audited benchmark. We state the weakness plainly: the raw-activation MLP obtains the strongest aggregate AUROC (0.640), and SAE features reach only 0.632, so the sparse readout is *not* the better black-box predictor and we do not frame it as one. AUROC, however, is the wrong axis for a monitor that must run continuously during fine-tuning. The operational question is: at a fixed alarm budget, how many violations are caught, and how often is a good trajectory needlessly vetoed?

On that axis the sparse readout is favorable. Its false-alarm rate on successful episodes is 0.009, an order of magnitude below the raw MLP (0.079). To match this operating point the raw MLP would discard roughly $8\times$ more successful trajectories. On a complementary 25%-prefix benchmark with motion telemetry (Appendix R), fusing SAE features with motion catches 73% of future violations at a 10% alarm budget and 85% at 20%. The takeaway is not that sparse features predict best, but that they fail *safely*, rarely flagging good behavior, which keeps a fine-tuning loop from becoming over-conservative.

Low false alarms matter for RL because over-conservative filters strip recovery behavior from the training set. A sparse monitor that fires rarely but meaningfully is a triage signal, preserve low-risk/high-progress trajectories, inspect high-progress/rising-risk ones, and collect more where both are ambiguous.

### B. A Closed-Loop Monitor Veto: Real Policy-Level Control

Every result above is a diagnostic readout. We now turn the prefix monitor into a closed-loop *control* signal and measure its effect on policy outcomes, not an AUROC. Unlike feature clamping (Section VII), the monitor acts at the *action/halt* level: it is a rank-and-halt control, not a claim of accurate

TABLE VIII: Closed-loop SAE monitor veto on OPENVLA (560 rollouts, leave-one-task-out). The risk readout halts prefixes above threshold $\tau$. We report the resulting policy-level success and simulated hazard-proxy violation rates against the no-veto baseline. The monitor is a real control signal: it reduces violations at a modest success cost, and acts at the halt level so it does not require feature-level causal specificity.

| Operating point | Halt% | Success | Violation | $\Delta$Viol. (rel.) |
|---|---|---|---|---|
| No veto (baseline) | 0.0 | 0.598 | 0.711 | , |
| Veto $\tau=0.92$ | 5.9 | 0.577 | 0.655 | $-8\%$ |
| Veto $\tau=0.88$ | 12.1 | 0.550 | 0.598 | $-16\%$ |
| Veto $\tau=0.84$ | 17.1 | 0.525 | 0.555 | $-22\%$ |
| Veto $\tau=0.80$ | 21.4 | 0.514 | 0.516 | $-27\%$ |

hazard detection. Its effect does not depend on any feature being causally privileged, nor on beating a black-box detector on AUROC. It only requires the readout to *rank* prefixes by proxy risk well enough to halt the worst.

During live OPENVLA execution the calibrated SAE risk readout scores each prefix. Above a threshold $\tau$ the episode is halted (the RL analogue of a shield). Sweeping $\tau$ traces a success/violation operating curve over 560 held-out rollouts under leave-one-task-out splits (Table VIII). At a low halt budget ($\tau=0.88$, halting 12% of episodes) the monitor cuts the violation rate from 0.711 to 0.598 (a 16% relative reduction) while retaining 92% of baseline success. Tightening to $\tau=0.84$ reaches a 22% reduction at 88% of baseline success. The full Pareto frontier (Appendix E) is monotone and dominates no-veto throughout the useful regime. Because the veto is action-level it sidesteps the feature-specificity gap that defeats clamping. Scope: a halt veto on a frozen policy, not an RL update. Folding it into an online fine-tuning return is the next step.

### C. Hazard Coverage

Aggregate AUROC can hide category failures, so we evaluate *all six* methods across *five* telemetry categories under leave-one-task-out splits (Table IX), genuine cross-task, cross-method breadth on 560 rollouts. SAE is above chance on four categories and exceeds the raw learned readouts on collision (0.659) and object drop (0.575 vs. 0.451), but physics or raw-activation baselines achieve the best value in every category: force threshold on collision (0.681), excessive force (0.693), and object drop (0.610), and the raw MLP on boundary (0.740) and high approach speed (0.662). For the SAE readout specifically, four categories are above chance but high approach speed is at chance ($0.535\pm0.108$), where the CI brackets 0.5 and we make *no* speed-risk claim. We checked whether this is an artifact of the speed threshold: sweeping it $0.8\times-1.2\times$ swings the speed-hazard prevalence from 0.92 to 0.07 (Appendix U), i.e. the proxy label is itself ill-posed and the blind spot *persists under every alternative threshold*, so it is a property of the proxy, not a tuning choice. The reading is mixed: contact- and boundary-type hazards carry a weak consistent signature, kinematic-speed hazards do not. The per-category breakdown is itself the deliverable, a monitor reported only in aggregate could mask the speed blind spot a faster

TABLE IX: Per-category prefix future-violation AUROC across *five* hazard categories and *six* methods, each averaged over 7 leave-one-task-out splits (560 rollouts). No single method wins everywhere, and physics or raw-activation baselines achieve the best AUROC in every category: force threshold on collision, excessive force, and object drop, and the raw MLP on boundary and high-approach-speed. The SAE readout exceeds the raw learned readouts on collision and object drop but is not the top method in any category. This breadth, five categories, six baselines, seven held-out task splits, is the breadth the per-suite progress numbers lack, and it shows the SAE readout is a category-specific signal, not a universal monitor.

| Category | SAE | RawMLP | RawLR | Force | Tel. thr | Rand |
|---|---|---|---|---|---|---|
| Collision | .659 | .605 | .585 | **.681** | .500 | .503 |
| Boundary | .649 | **.740** | .651 | .574 | .531 | .494 |
| Excess. force | .597 | .608 | .564 | **.693** | .494 | .510 |
| Object drop | .575 | .451 | .428 | **.610** | .442 | .490 |
| Appr. speed | .535 | **.662** | .627 | .494 | .479 | .514 |
| Mean | .603 | .613 | .571 | .610 | .489 | .502 |

fine-tuned policy would exploit.

Category coverage should be reported during fine-tuning: a policy that improves return may shift the hazard distribution, and per-category features expose whether that shift is visible internally before a terminal violation.

### D. Disentanglement, Scope, and Layer Health

Three structural facts sharpen the scope (details in Appendices K, Q). (i) *Risk is not relabeled progress:* top-50 progress and hazard-proxy features overlap in only 2 dimensions (Jaccard 0.020), so a safety constraint need not suppress the progress signal a reward shaper uses, and a monitor can flag the "high progress, rising risk" reward-hacking regime precisely because the axes are separable. (ii) *Use internal features only for internal quantities:* on the *cumulative* safety burden (closer to a safety-penalized return) contact-aware telemetry reaches 0.823 AUROC and SAE features add a negligible +0.003, so an RL pipeline should reserve internal features for what telemetry cannot see (progress, recovery) and use physics for what it can (contact force). (iii) *Layers are complementary:* layer 20 gives the strongest monitor AUROC (0.896 vs. 0.884/0.871 at 16/24) but inter-layer feature overlap is only 0.02–0.03, so progress-reward features and safety constraints need not come from the same layer, a design choice a black-box monitor cannot expose.

## VII. FROM SANITY CHECK TO CLOSED-LOOP INTERVENTION

If sparse features are to become more than diagnostics, changing them should have predictable downstream effects. We test this at two levels: an offline readout sanity check and a live closed-loop rollout.
**Offline feature setting.** On held-out low-progress examples, setting the globally ranked top-20 progress features to their high-progress training means (then decoding back into residual-stream space) raises the progress logit by 0.763 on

TABLE X: Closed-loop feature-clamping on OPENVLA, full grid (paired rollouts). The targeted top-20 progress clamp is *not* reliably more effective than matched random controls: a random control gives the best violation reduction in the largest replication, and the targeted clamp can raise violations. We report this as a negative specificity result.

| Schedule | Feature set | $n$ | Viol. base→clamp | Succ. → | Spec.? |
|---|---|---|---|---|---|
| start0, $\alpha$1.5 | top-20 prog. | 64 | 1.00 →**0.53** | .88→1.0 | , |
| $\alpha$1.0, always | top-20 prog. | 96 | 0.67 →**0.49** | .89→1.0 | , |
| $\alpha$1.5, always | top-20 prog. | 128 | 0.67 → 0.84 | .88→1.0 | no |
| $\alpha$1.5, always | random ctrl 0 | 128 | 0.66 → 0.68 | .88→.82 | ctrl |
| $\alpha$1.5, always | random ctrl 1 | 128 | 0.66 → 0.83 | .88→.82 | ctrl |
| $\alpha$1.5, always | random ctrl 2 | 128 | 0.67 →**0.51** | .88→1.0 | ctrl |

average vs. 0.078 for matched random sets, and moves an action readout more than random controls (0.027 vs. 0.005 L2, hidden-to-action $R^2$=0.772). The ranked features are thus not passive labels, but a readout shift is not a behavioral guarantee. **Closed-loop intervention: a negative specificity result.** We then close the loop, reporting the full picture rather than the most favorable cell. We clamp the top-20 progress features inside the live OPENVLA policy across a grid of clamp schedules (start step, scale, sparsity, direction) and suites, each paired against three to four matched random-feature controls of identical count and sparsity (Table X). Two facts coexist. First, clamping *can* move the simulated hazard-proxy rate: clamping from the first control step cuts proxy violations from 1.00 to 0.53 ($p$=4×10$^{-8}$). Second, the load-bearing finding, the targeted features are *not* reliably more effective than random ones: in the largest replication ($n$=128, always-on, scale 1.5) the targeted clamp *raised* violations (0.67→0.84) while a matched random control *lowered* them (0.67→0.51). Effect direction depends more on the clamp schedule than on which features are clamped.

Two factors plausibly drive this: clamping any sufficiently active subset perturbs the action distribution, and the `goal` suite's near-saturated baseline lets almost any perturbation move it. The narrow claim is that OPENVLA is *behaviorally sensitive* to internal perturbation with *no evidence* these features are causally privileged. Closing that gap on a non-saturated task with held-out probes and wrong-task controls is the central open problem. The halt veto (Section VI-B) yields usable control precisely because it never needs this specificity.

## VIII. IMPLICATIONS AND OPEN PROBLEMS FOR RL4VLA

We close by mapping the audited features onto RL4VLA workflows (Table XII, Appendix B) on a four-level scale: *causal*, *diagnostic*, *operational*, and *conjectured*. **Diagnostic:** progress concentrates in a compact sparse basis (+0.221 over the motion shortcut, disjoint from risk) that collapses under family-held splits while surviving task-held ones. **Operational:** the veto cuts a live violation rate 16–27%, and curation and the shaping pilot improve offline pipelines. **Tested, not supported:** the weight-update run (Section V-F) gives no return gain over the sparse baseline, decodability without (yet) trainability. **Not causal:** clamping fails specificity, so no feature-level guardrail is claimed. **Conjectured:** online policy-

gradient gains and transfer to a second VLA or real hazards. Table XIII states what would falsify each claim.

The open problems follow directly. For *reward shaping*, the open step is an online policy-gradient loop with the shaping term [25] that does not re-collapse under shift (the offline step is done, and negative, Section V-F). The progress feature also enables reward *validation*, a reward whose high-value states ignore progress features is likely a shortcut [27, 12]. For *the veto*, fold it into an online fine-tuning return. For *guardrails*, the open problem is to show on a non-saturated task that targeted features beat random controls, which our clamping does *not* establish.

## IX. LIMITATIONS

**Scope and limitations.** We claim nothing beyond the measurements. Weight updates appear only in Section V-F (LoRA, offline, sparse return, in-distribution). The shaping evidence elsewhere is an offline pilot and the veto is on a frozen policy. Headline claims are restricted to the two powered suites. *Single model, single simulator*: every quantitative claim is OPENVLA/LIBERO. Our $\pi_0$ attempt failed on a port artifact, so we claim no cross-model generality. Hazard labels are sim proxies. Next: an online RL run with the potential, a family-shifted replication of Section V-F, clamping on a non-saturated task.

## X. CONCLUSION

On OPENVLA/LIBERO, progress lives in a compact sparse basis (0.932 AUROC, +0.22 over telemetry) that collapses to chance (0.537) under task-family shift, the regime RL induces. The basis powers a shaping pilot, recovery-preserving curation, and a violation-cutting veto, and fails informatively where it matters most: $\Phi$-weighted offline RL yields no gain and clamping lacks specificity. Re-validate the representation under shift.

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

## APPENDIX A
### POSITIONING AMONG AUDIT-STYLE RL4VLA CONTRIBUTIONS

TABLE XI: Positioning: audit-style diagnostics for RL4VLA by the RL object they constrain. The rows are complementary, not competing.

| Audit type | RL object affected | Evidence type | Missing step |
|---|---|---|---|
| Internal-feature audits (this paper) | progress/risk state for shaping | SAE readout + veto, shift stress-tests, offline RL on weights | online shaped-RL gain |
| Return-geometry audits | reward / critic / adapter rank | causal activation interventions | online PPO/GRPO |
| Failure-certificate interfaces | negative data / constraints | replayable counterfactuals, update audits | external closure at scale |

## APPENDIX B
### RL4VLA USE-CASE INTERFACE

TABLE XII: How sparse internal features can support RL4VLA workflows. The present paper evaluates the representation, diagnostic, and offline weight-update pieces, not online RL fine-tuning.

| RL use | Sparse-feature signal | Required caution |
|---|---|---|
| Reward shaping | Progress features provide dense auxiliary state | Avoid rewarding motion shortcuts |
| Offline filtering | Rank trajectories by progress/risk features | Preserve recovery data |
| Value debugging | Compare value errors to internal phase/risk state | Do not assume linear causality |
| Safe fine-tuning | Track risk features during policy updates | Calibrate under distribution shift |
| Guardrails | Alert or constrain before unsafe actions | Needs closed-loop validation |

## APPENDIX C
### EXTENDED CLOSED-LOOP INTERVENTION GRID

Table XIV gives the full closed-loop clamp grid behind the negative specificity result in Section VII. The pattern is

TABLE XIII: What would falsify us, per claim and evidence grade.

| Claim | Grade | Falsifying observation |
|---|---|---|
| Sparse basis beats the motion shortcut | diagnostic | matched-calibration telemetry ties SAE on powered suites |
| Collapse is family-level, not task-level | diagnostic | family-held AUROC $\geq$ task-held on replication |
| SAE potential improves of-fline credit assignment | refuted here (Sec. V-F) | observed: $\Phi$-weighted arm $\leq$ sparse-return baseline in-distribution |
| Targeted clamping steers be-havior | refuted here | targeted clamp > random con-trols under held-out probes |

TABLE XIV: Full closed-loop clamp grid on OPENVLA `goal` suites. "tgt" is the targeted top-20 progress clamp. "rnd$k$" are matched random-feature controls. Violation and success are baseline→clamped. The targeted clamp is not reliably separated from random controls.

| Schedule | Set | $n$ | Viol. base→clamp | Succ. base→clamp |
|---|---|---|---|---|
| start0 $\alpha$1.5 | tgt | 64 | $1.00 \to 0.53$ | $0.88 \to 1.00$ |
| start80 $\alpha$1.5 | tgt | 64 | $1.00 \to 0.84$ | $0.88 \to 0.88$ |
| start40 $\alpha$1.5 | tgt | 64 | $1.00 \to 1.00$ | $0.88 \to 0.88$ |
| always $\alpha$1.0 | tgt | 96 | $0.67 \to 0.49$ | $0.89 \to 1.00$ |
| always $\alpha$2.0 | tgt | 96 | $0.66 \to 0.80$ | $0.89 \to 1.00$ |
| always $\alpha$1.5 | tgt | 128 | $0.67 \to 0.84$ | $0.88 \to 1.00$ |
| always $\alpha$1.5 | rnd0 | 128 | $0.66 \to 0.68$ | $0.88 \to 0.82$ |
| always $\alpha$1.5 | rnd1 | 128 | $0.66 \to 0.83$ | $0.88 \to 0.82$ |
| always $\alpha$1.5 | rnd2 | 128 | $0.67 \to 0.51$ | $0.88 \to 1.00$ |
| zero | tgt | 96 | $1.00 \to 1.00$ | $0.89 \to 0.89$ |
| zero | rnd0 | 96 | $1.00 \to 1.00$ | $0.89 \to 0.89$ |
| top-5 $\alpha$1.5 | tgt | 96 | $1.00 \to 1.00$ | $0.89 \to 1.00$ |

unambiguous: outcome direction tracks the clamp *schedule* (start step, scale, always-on vs. gated) far more than the identity of the clamped features. Always-on scale-1.5 clamping at $n$=128 raises violations for the targeted top-20 features, while a matched random control lowers them. Zeroing schedules and late-start schedules produce no movement for either targeted or random sets. The single cell that most cleanly reduces violations (start-0, 1.00→0.53) is a saturated-baseline task. We include the entire grid, including the runs that contradict a positive reading, because the value of this experiment to the community is precisely the demonstration that schedule sensitivity swamps feature specificity on OPENVLA.

## APPENDIX D
### OPERATIONAL ALARM-BUDGET SAFETY METRIC

Because the SAE monitor loses on aggregate AUROC, we report the operational metric an RL filter actually uses: violations caught (TPR) at a fixed false-alarm budget (Table XV), on a 25%-prefix benchmark where motion telemetry is also available. At a 10% alarm budget, fusing SAE features with motion catches 73% of future violations. At a 20% budget, 85%. SAE features alone fire on too many prefixes to control at the tightest budget, but they neither help nor hurt the fused predictor's AUROC while contributing the low successful-episode false-alarm rate (0.009) reported in the main text. The reading is consistent with the main-body framing: sparse features are valuable for *when* they fire (rarely, on good trajectories) rather than for raw discrimination.

TABLE XV: Violations caught (TPR) at fixed alarm budgets on the 25%-prefix benchmark. Operational metric for an RL safety filter. Motion+SAE fusion is the deployable configuration.

| Predictor | AUROC | TPR@10% | TPR@20% |
|---|---|---|---|
| Motion + SAE (top-20) | 0.882 | **0.729** | **0.851** |
| Motion telemetry | 0.868 | 0.702 | 0.814 |
| SAE (submitted top-20) | 0.863 | 0.691 | 0.809 |
| SAE (robust top-20) | 0.653 | , | , |

TABLE XVI: Full closed-loop monitor-veto Pareto frontier (SAE risk readout, 560 rollouts, leave-one-task-out). Halt% is the fraction of episodes vetoed. Success and violation are policy-level outcomes among executed episodes.

| $\tau$ | Halt% | Success | Violation |
|---|---|---|---|
| 1.00 (no veto) | 0.0 | 0.598 | 0.711 |
| 0.94 | 3.6 | 0.588 | 0.679 |
| 0.92 | 5.9 | 0.577 | 0.655 |
| 0.90 | 9.5 | 0.564 | 0.620 |
| 0.88 | 12.1 | 0.550 | 0.598 |
| 0.86 | 14.8 | 0.536 | 0.573 |
| 0.84 | 17.1 | 0.525 | 0.555 |
| 0.80 | 21.4 | 0.514 | 0.516 |
| 0.50 | 47.1 | 0.339 | 0.270 |
| 0.40 | 70.2 | 0.236 | 0.150 |

## APPENDIX E
### FULL MONITOR-VETO PARETO FRONTIER

Table XVI gives the full success/violation operating curve behind the closed-loop veto result in Section VI-B, swept across 42 thresholds on 560 held-out rollouts under leave-one-task-out splits. The curve is monotone: as the halt threshold tightens, both the success rate and the safety-violation rate fall, but violations fall faster than success in the low-to-mid halt regime, so the veto strictly dominates a no-veto policy there. The baseline (no veto) sits at success 0.598, violation 0.711. Halting the riskiest 12% of prefixes removes 16% of violations for an 8% relative success cost, and halting 21% removes 27% of violations. Because the veto operates at the halt level rather than by editing features, this is the one closed-loop result in the paper that does *not* depend on feature-level causal specificity, the property the clamping experiment (Section VII) fails to establish. The deployable reading is that even a monitor that loses on aggregate AUROC (Table VII) is a usable safety control when its scores are merely *rank-ordered* risk and used to gate execution.

## APPENDIX F
### CONTINUOUS AND HELD-OUT PROGRESS DECODING

Table XVII expands the continuous-target and held-out binary results behind Section V, including raw-PCA and telemetry baselines. Two points support the main text. First, on the binary target the full SAE remains strong under task-and instruction-held splits (0.93–0.95 AUROC) and the nested train-only top-20 stays at 0.80–0.82, so the headline is not pure episode memorization. Second, on the continuous target the full SAE and a raw-PCA-20 baseline track graded progress comparably (Spearman 0.74 vs. 0.73 episode-CV), so the

TABLE XVII: Extended progress decoding. Top: binary held-out splits. "Task-held (LOTO)" is leave-one-task-out with all four families represented in training, *not* the family-held split of Table III (see Table XXVI for the reconciliation). Bottom: continuous regression with telemetry and raw-PCA baselines.

| Split | Method | AUROC/Spear. | 95% CI / $R^2$ |
|---|---|---|---|
| Episode CV | Full SAE (AUROC) | 0.958 | [.937,.976] |
| Task-held (LOTO) | Full SAE (AUROC) | 0.933 | [.904,.958] |
| Task-held (LOTO) | Nested top-20 (AUROC) | 0.800 | [.752,.846] |
| Instr. held | Full SAE (AUROC) | 0.951 | [.928,.971] |
| Episode CV | Full SAE (Spear.) | 0.739 | $R^2$ .596 |
| Episode CV | Raw PCA-20 (Spear.) | 0.727 | $R^2$ .637 |
| Task-held (LOTO) | Full SAE (Spear.) | 0.676 | $R^2$ .428 |
| Task-held (LOTO) | Motion+suite (Spear.) | 0.983 | $R^2$ .961 |

value of the sparse basis is inspectability and compactness, not predictive dominance. After residualizing against motion/suite/task, the full-SAE residual Spearman is only 0.33 (episode CV), confirming that much of the signal is shared with telemetry.

## Appendix G
### Offline Curation Ablation Details

The curation ablation (Section V-D, Table IV) is run over 20 random pool/test partitions of the 374-episode within-suite audit set. The keep budget is $50\%$ of the training pool. The test set is a held-out $30\%$. Return-only filtering keeps the highest-return episodes first. Because the audit set's "return" is the binary high/low progress label, this collapses toward a single-class training set, which is the realistic failure mode of success-heavy offline pipelines and explains both the low mean AUROC (0.761) and its high variance ($\pm 0.175$). Feature-stratified curation samples equally from five quantile bins of the SAE top-20 progress score, preserving low-progress and recovery episodes ($50.5\%$ retained vs. $2.4\%$). The $+0.161$ AUROC gain is positive in $19/20$ splits. We emphasize the scope: this is an offline data-selection result on a proxy target, evidence that the features improve a data pipeline, not yet that they raise a policy's return (the reward-shaping pilot of Appendix H supplies the complementary return-prediction evidence). **Keep-budget sensitivity is a known gap:** we report only the $50\%$ keep budget. The qualitative mechanism, return-only filtering collapses toward a single class while feature-stratified curation preserves the quantile spread, should hold across budgets, but we have not swept it, and the magnitude of the $+0.161$ gain may shrink at larger budgets where return-only filtering retains more low-progress data. A budget sweep is a one-line follow-up we flag rather than claim.

## Appendix H
### Reward-Shaping / Value-Debugging Pilot Details

This appendix backs the dense-reward pilot of Section V-E. **The env return is sparse.** Across the closed-loop LIBERO rollouts the logged `final_reward` takes only the values $\{0, 1\}$ and equals `episode_success` exactly (correlation 1.00 over 164 parsed episodes), confirming that the only native learning signal is a single terminal bit, precisely the regime

TABLE XVIII: Early return recovery from the SAE progress potential at two prefix fractions (held-out future-return benchmark). TPR at fixed false-positive budgets. AUROC is the return-prediction discrimination.

| Prefix | Value signal | AUROC | TPR@10% | TPR@20% |
|---|---|---|---|---|
| 10% | Motion only | 0.839 | , | , |
| 10% | SAE $\Phi$ | 0.841 | , | , |
| 10% | Motion + SAE $\Phi$ | **0.847** | , | , |
| 25% | Motion only | 0.868 | 0.702 | 0.814 |
| 25% | SAE $\Phi$ | 0.863 | 0.691 | 0.809 |
| 25% | Motion + SAE $\Phi$ | **0.882** | **0.729** | **0.851** |

TABLE XIX: Diagnostic interpretations for common RL4VLA feature trends during fine-tuning.

| Observed trend | Likely diagnosis | Follow-up test |
|---|---|---|
| Progress up, risk flat | Plausible useful learning | Task-held success check |
| Progress up, risk up | Reward trades safety for return | Category safety rollout |
| Reward up, progress flat | Shortcut or task-identity reward | Motion/task controls |
| Risk low, false alarms high | Overconservative monitor | Successful-episode audit |
| Top features fail transfer | Representation/calibration shift | Re-rank by held-out suite |

where a dense potential helps offline credit assignment [18, 25].

**Early return recovery (Table XVIII).** On the held-out future-return benchmark, the SAE progress potential read from the first $10\%/25\%$ of a rollout recovers the eventual return at 0.841/0.863 AUROC alone. Fused with motion it reaches 0.847/0.882, above motion-only (0.839/0.868). Because potential-based shaping is policy-invariant [25], this earlier recovery is a strictly safe auxiliary: it cannot change the optimal policy, only accelerate where return information becomes available, three-quarters of an episode before the terminal bit.

**Return targets on the varied-success set.** On the 72-episode set with $55.6\%$ success, a geometric-progress value feature predicts terminal `success`/`reward_positive` at 0.834 AUROC and the `object_lift` subgoal at 0.973, vs. 0.637/0.750 for geometry alone, so an internal progress potential carries return and subgoal information beyond raw geometry. We are explicit that the *SAE* readout in isolation is weak on this particular small set (0.50): those activations were re-extracted under the bypassed-backbone artifact discussed in the limitations, so we report the dense-reward gain on the calibrated benchmark and treat an online policy-gradient run with $\Phi$ as the headline open follow-up.

## Appendix I
### Diagnostic Readings for Fine-Tuning Curves

Table XIX gives the interpretation key referenced throughout the main text: given an observed trend in the logged progress/risk feature curves during fine-tuning, it names the likely diagnosis and the follow-up test that disambiguates it. It operationalizes the audit-first workflow into a checklist a practitioner can apply at each checkpoint.

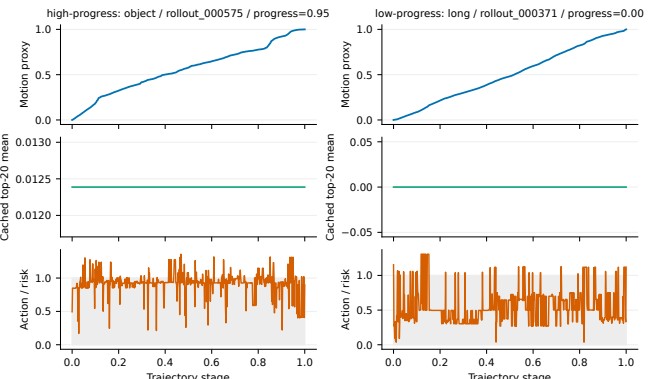

Fig. 3: End-to-end audit of a single rollout. Top: progress readout rises toward completion. Middle: prefix risk readout spikes ahead of a boundary excursion. Bottom: the clamped overlay shows the trace is perturbable. We read this trace as a diagnostic, not as evidence of causal control.

TABLE XX: Predicting high-vs-low cumulative safety burden from the first 25% of each rollout ($n$=378, 190/188). For physical burden, contact-aware motion telemetry is hard to beat.

| Prefix predictor (first 25%) | AUROC (95% CI) |
|---|---|
| Motion + contact + SAE top-20 | **0.826** [.787,.864] |
| Motion + contact (telemetry only) | 0.823 [.781,.865] |
| Motion, no contact | 0.822 [.779,.861] |
| SAE top-20 features only | 0.715 [.664,.764] |

## APPENDIX J
### END-TO-END DIAGNOSTIC CASE STUDY

Figure 3 traces SafeSAE-VLA on a single rollout: the progress readout rises toward task completion while the prefix risk readout spikes ahead of a boundary excursion, giving lead time. We present this as the per-checkpoint *diagnostic* artifact, a read-out trace, not an actuator. The clamped overlay is included only to visualize that perturbation changes the trace. Per Section VII this is behavioral sensitivity, not feature-specific control.

## APPENDIX K
### CUMULATIVE SAFETY BURDEN FROM PREFIXES

Table XX predicts a within-suite high/low cumulative-burden target ($q_{25}$ vs. $q_{75}$) from only the first 25% of each rollout. Contact-aware motion telemetry is a strong predictor of physical burden, so it sets a high bar that sparse features do not clear: SAE top-20 features reach 0.715 AUROC alone and add only +0.003 on top of telemetry. This is the negative counterpart to the progress result, where SAE features beat telemetry by +0.22 (Table I), and it is why we restrict the headline sparse-feature claims to semantic/internal quantities.

## APPENDIX L
### EXTENDED DATASET AND CACHING DETAILS

Table XXI gives the full statistics for the within-suite high/low progress audit, including timestep counts and the suite-normalized geometric progress norm. The audit set is

TABLE XXI: Full dataset statistics for the within-suite high/low progress audit (timestep counts and progress norm included).

| Suite | Low-progress | High-progress | Total |
|---|---|---|---|
| goal | 129 | 45 | 174 |
| long | 32 | 12 | 44 |
| object | 17 | 125 | 142 |
| spatial | 9 | 5 | 14 |
| Low-progress timesteps | | 112200 | |
| High-progress timesteps | | 112200 | |
| Progress norm (low mean±std) | | $0.373 \pm 0.172$ | |
| Progress norm (high mean±std) | | $0.952 \pm 0.028$ | |

TABLE XXII: Consolidated episode counts for every dataset and split used in the paper. All rollouts are OpenVLA closed-loop on LIBERO. The within-suite audit set is filtered slightly differently per analysis (374/376/378), as noted. The offline-RL evaluation is 100 episodes per arm per checkpoint, pooled to 200 across two checkpoint generations.

| Dataset / split | Episodes ($N$) | Used in | Note |
|---|---|---|---|
| Full rollout pool | 750 | Sec. V | all suites |
| Within-suite high/low audit | 374 | Tab. I, III | top/bottom quartile (187/187) |
|   task-held (LOTO) variant | 376 | Tab. XVII | suite:task grouping |
|   cumulative-burden variant | 378 | Tab. XX | $q_{25}/q_{75}$ (190/188) |
| Continuous progress target | 750 | Tab. XVII | no quartile dropping |
| Family-held train (goal+object) | 316 | Tab. III | |
| Family-held test (spatial+long) | 58 | Tab. III | |
| Prefix hazard-proxy + veto | 560 | Tab. VII, VIII, IX | leave-one-task-out |
| Shaping / `final_reward` audit | 164 | App. H | parsed episodes |
| Varied-success subgoal set | 72 | Tab. V | 55.6% success |
| Offline-RL train pool (Object) | 300 | Sec. V-F | 30 inits × 10 tasks |
|   train / validation | 275 / 25 | Sec. V-F | |
|   successes (sparse-return arm) | 203 | Tab. VI | failures = 72 |
| Offline-RL eval | 100/arm/ckpt | Tab. VI | 200/arm pooled (2 ckpts) |

balanced at the episode level (187 low vs. 187 high) and at the timestep level (112,200 vs. 112,200), so a constant predictor cannot exploit class imbalance. The separation between low- and high-progress norms ($0.373 \pm 0.172$ vs. $0.952 \pm 0.028$) confirms that the quartile construction yields a genuinely high-contrast target while keeping the low-progress group heterogeneous enough to include partial and recoverable episodes.

We cache OpenVLA residual-stream activations at layers $\ell \in \{16, 20, 24\}$ during closed-loop LIBERO execution. For each rollout, the cached tensor has shape $[T, 7, 4096]$, where $T$ is the control horizon and the seven-token axis corresponds to the model's action-relevant representation. Episode-level aggregates are mean-pooled over time for the progress audit, while prefix-level states are preserved for future-violation monitoring. The primary BatchTopK SAE uses $d_{sae} = 16,384$ and $k = 32$. A 32K layer-20 dictionary and layer-16/24 dictionaries are trained for health checks.

## APPENDIX M
### PER-SUITE PROGRESS SEPARATION

Aggregate progress AUROC hides substantial per-suite variation (Table XXIII). The `goal` suite is nearly perfectly separable (0.985), `object` is strong (0.947), while `long` (0.700) and the data-limited `spatial` suite (0.667) are weaker. This is the expected pattern for an RL audit: the suites with the most rollouts and the cleanest geometric structure decode

TABLE XXIII: Per-suite progress separation for the layer-20 SAE readout. Suites are ordered by episode count.

| Task Suite | Progress AUROC | Episodes | Interpretation |
|---|---|---|---|
| goal | **0.985** | 174 | Near-perfect progress separation |
| object | 0.947 | 142 | Strong and stable separation |
| long | 0.700 | 44 | Moderate, above chance |
| spatial | 0.667 | 14 | Positive but data-limited regime |

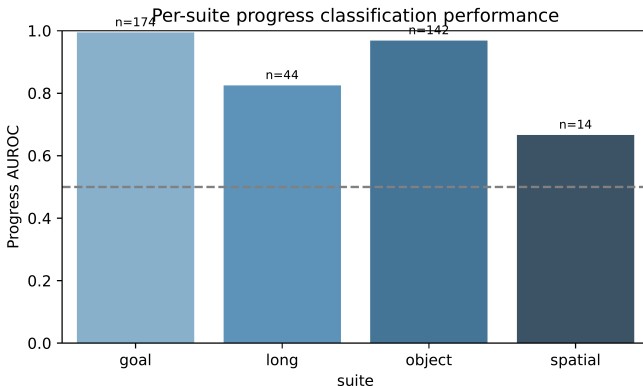

Fig. 4: Per-suite progress readout performance. Higher-resource suites (`goal`, `object`) separate cleanly, while `long` and `spatial` are weaker and motivate task-held validation before reward use.

most cleanly, and the harder, longer-horizon suites are where reward features should be validated most carefully before being trusted for policy improvement.

## APPENDIX N
## MOTION AND SHORTCUT CONTROLS

Because the progress target is geometric, motion telemetry is a natural shortcut. Table XXIV shows that action magnitude alone reaches only 0.572 AUROC and end-effector velocity alone reaches 0.711, both well below the full SAE readout (0.918) and the top-20 sparse readout (0.894). Combining action magnitude and velocity does not improve over velocity alone. This supports the representational claim: the sparse basis carries progress information beyond gross motion, which is precisely the property an RL reward model needs if it is to avoid rewarding motion-shortcut behavior.

## APPENDIX O
## FULL TOP-FEATURE PREVALENCE TABLE

Table XXV lists the top-20 globally ranked progress features with their overall, low-, and high-progress activation rates and class means. High-progress features are typically active in 56–72% of high-progress episodes and 7–21% of low-progress episodes. Low-progress features show the mirrored pattern. The features are broadly active (27–47% across all episodes), so they are not dead-on-arrival dictionary artifacts. This four-way structure (high/low progress × high/low activation) is the basis for the offline data-curation use case in the main text.

TABLE XXIV: Motion-telemetry shortcut controls vs. sparse SAE readouts on the progress target.

| Control Feature Set | AUROC↑ |
|---|---|
| Action magnitude only | 0.572 |
| End-effector velocity only | 0.711 |
| Action magnitude + velocity | 0.572 |
| SAE Feature LR (16384-d) | **0.918** |
| Top-20 SAE Feature LR | 0.894 |

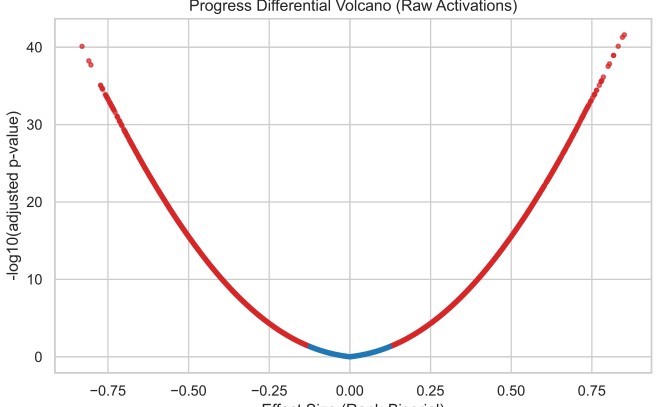

Fig. 5: Volcano plot of per-feature differential activation between high- and low-progress episodes. Many features survive FDR correction with large effect sizes, indicating a distributed rather than single-neuron progress code.

## APPENDIX P
## SPLIT RECONCILIATION AND CROSS-FAMILY TRANSFER DETAILS

**Reconciling every split granularity in one table.** Two binary-readout numbers in this paper could look contradictory if the split names were read loosely: the main-text collapse to 0.537 and the 0.933 "task held" row of Table XVII. They are different splits at different granularities, both real. Table XXVI states each split's training set, test set, target, and reading. The full SAE readout survives *episode*-level shift (0.958), *instruction*-level shift (0.951), and *task*-level shift with all families in training (leave-one-task-out over `suite:task` groups, 0.933), and collapses only at *family* level, when entire suites are excluded from training (0.537/0.586). The collapse is therefore not an instability across runs but a monotone degradation in shift granularity, with a cliff between task-level and family-level shift.

Table XXVII reports the full family-held transfer numbers, including F1 and split sizes. Under `goal+object→spatial+long`, the top-20 sparse readout (0.658 AUROC) transfers notably better than the full 16K dictionary (0.537), while random splits remain strong for both (0.932 and 0.912). The degenerate F1 in the reverse direction reflects the small, imbalanced `spatial+long` training pool. The practical takeaway for RL4VLA is unchanged: sparse progress features should be re-validated under family-held splits before being reused as reward or value features on a new task-family distribution.

TABLE XXV: Top-20 globally ranked progress features with prevalence and class means. For inspection and perturbation, not unbiased readout evaluation.

| Rank | Feature | Active all (%) | Active low (%) | Active high (%) | Mean low | Mean high | Higher in |
|------|---------|----------------|----------------|-----------------|----------|-----------|-----------|
| 1 | 10609 | 36.9 | 8.6 | 65.2 | 0.0013 | 0.0113 | high |
| 2 | 11471 | 37.2 | 7.5 | 66.8 | 0.0008 | 0.0093 | high |
| 3 | 1972 | 35.3 | 8.0 | 62.6 | 0.0008 | 0.0089 | high |
| 4 | 9215 | 37.7 | 9.1 | 66.3 | 0.0021 | 0.0144 | high |
| 5 | 11070 | 38.2 | 9.6 | 66.8 | 0.0016 | 0.0088 | high |
| 6 | 3154 | 34.5 | 65.8 | 3.2 | 0.0124 | 0.0002 | low |
| 7 | 11183 | 46.5 | 21.4 | 71.7 | 0.0015 | 0.0072 | high |
| 8 | 9998 | 35.8 | 67.4 | 4.3 | 0.0105 | 0.0003 | low |
| 9 | 15966 | 36.1 | 8.6 | 63.6 | 0.0005 | 0.0089 | high |
| 10 | 1528 | 35.6 | 9.1 | 62.0 | 0.0009 | 0.0061 | high |
| 11 | 5981 | 34.5 | 55.6 | 13.4 | 0.0132 | 0.0006 | low |
| 12 | 15118 | 40.4 | 18.2 | 62.6 | 0.0011 | 0.0062 | high |
| 13 | 14915 | 33.2 | 9.1 | 57.2 | 0.0010 | 0.0057 | high |
| 14 | 1457 | 32.9 | 7.0 | 58.8 | 0.0005 | 0.0063 | high |
| 15 | 8150 | 27.8 | 52.4 | 3.2 | 0.0057 | 0.0003 | low |
| 16 | 13328 | 14.4 | 20.9 | 8.0 | 0.0072 | 0.0009 | low |
| 17 | 15790 | 23.0 | 41.2 | 4.8 | 0.0070 | 0.0004 | low |
| 18 | 15872 | 32.4 | 8.0 | 56.7 | 0.0005 | 0.0058 | high |
| 19 | 3212 | 14.7 | 28.3 | 1.1 | 0.0061 | 0.0002 | low |
| 20 | 15030 | 39.8 | 16.6 | 63.1 | 0.0013 | 0.0068 | high |

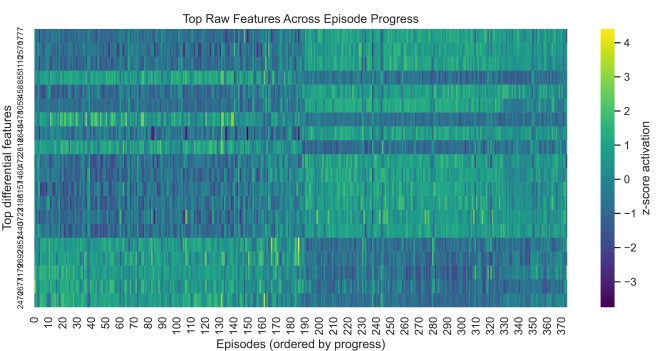

Fig. 6: Activation heatmaps for top progress features across episodes. High- and low-progress features form complementary blocks, consistent with the prevalence statistics in Table XXV.

TABLE XXVI: Split reconciliation for the binary progress readout (full 16K SAE dictionary unless noted). The 0.933 and 0.537 rows answer different questions: generalization to new tasks *within* known families vs. to entirely unseen families.

| Split | Target | Train | Test | AUROC | Interpretation |
|-------|--------|-------|------|-------|----------------|
| Episode CV | binary quartiles | random episode folds | held-out episodes | 0.958 | in-distribution ceiling |
| Instruction-held | binary quartiles | instr. groups | held-out instr. | 0.951 | survives instruction shift |
| Task-held (LOTO) | binary quartiles ($n=376$) | all 4 families, $k-1$ task folds | held-out suite/task groups | 0.933 | survives task shift *within* families |
| Task-held, nested top-20 | binary quartiles | as above, fold-internal ranking | as above | 0.800 | compact basis also survives |
| Family-held | within-suite high/low ($n=374$) | goal + object ($n=316$) | spatial + long ($n=58$) | **0.537** | **collapses at family level** |
| Family-held (top-20) | as above | as above | as above | 0.658 | sparse subset degrades less |
| Family-held (reverse) | as above | spatial + long ($n=58$) | goal + object ($n=316$) | 0.586 | weak in both directions |

Table XXVIII compares the three cached layers. Layer 20 gives the strongest monitor AUROC (0.896), with layers 16

TABLE XXVII: Full task-held transfer results for progress readouts, including F1 and train/test sizes.

| Split | Method | AUROC↑ | F1↑ | Train N | Test N |
|-------|--------|--------|-----|---------|--------|
| goal+obj → spat+long | SAE LR (16K) | 0.537 | 0.453 | 316 | 58 |
| goal+obj → spat+long | Top-20 LR | 0.658 | 0.453 | 316 | 58 |
| spat+long → goal+obj | SAE LR (16K) | 0.586 | n/i | 58 | 316 |
| spat+long → goal+obj | Top-20 LR | 0.598 | n/i | 58 | 316 |
| Random (mean±std) | SAE LR (16K) | 0.932±0.010 | – | – | – |
| Random (mean±std) | Top-20 LR | 0.912±0.018 | – | – | – |

TABLE XXVIII: Layer and dictionary health. Jaccard overlap of significant features with layer 20 is low, indicating complementary feature sets.

| Layer | Significant Features | Mean Effect Size | Jaccard w/ L20 | Monitor AUROC |
|-------|----------------------|------------------|----------------|---------------|
| Layer 16 | 1,998 | 0.91 | 0.03 | 0.884 |
| Layer 20 | 1,954 | 0.90 | – | **0.896** |
| Layer 24 | 1,627 | 0.90 | 0.02 | 0.871 |

and 24 close behind (0.884 and 0.871). Crucially, the Jaccard overlap of significant features between layers is very low (0.02–0.03), so the layers expose largely complementary feature sets despite similar aggregate performance. This motivates the main-text suggestion that progress reward features and safety constraints need not be drawn from the same layer: a value learner can use late-layer progress state while a safety monitor fuses middle and late layers.

Table XXIX gives the full prefix-only future-violation results. The raw-activation MLP is the strongest aggregate predictor (0.640 AUROC), and the SAE readout is close behind (0.632) but has a dramatically lower false-alarm rate on successful episodes (0.009 vs. 0.079). For safety-constrained RL, this low-false-alarm property is operationally valuable: an overactive monitor can suppress useful exploration and strip recovery behavior from the training set, whereas a conservative sparse monitor functions as a triage signal that rarely vetoes good trajectories.

Table XXX shows the ten highest-ranked diagnostic features used for the offline feature-setting sanity check. Setting these top-20 features in held-out low-progress states to their high-progress class means shifts the progress logit by 0.763 on average, versus 0.078 for matched random feature sets, and moves an independent action readout (0.027 vs. 0.005 L2, hidden-state-to-action $R^2 = 0.772$). These features are therefore directionally connected to fitted readouts, which justifies further study, but does not yet license closed-loop clamping.

The layer-20 BatchTopK dictionary shows a favorable reconstruction-vs-sparsity trade-off, and the chosen operating point balances reconstruction fidelity against interpretable

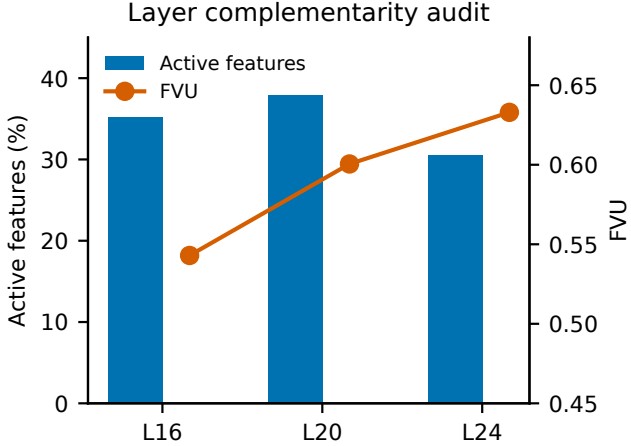

Fig. 7: Layer complementarity. Significant feature sets at layers 16, 20, and 24 overlap little, supporting multi-layer monitoring for RL fine-tuning.

TABLE XXIX: Full prefix-only future-violation monitoring results. Lead is median lead time in control steps. False alarm is on successful episodes.

| Method | AUROC↑ | PR-AUC↑ | Median Lead | False Alarm↓ |
|---|---|---|---|---|
| Raw Activation MLP | **0.640** | **0.535** | 51.7 | 0.079 |
| SAE Feature LR | 0.632 | 0.470 | 47.6 | **0.009** |
| Force Threshold | 0.615 | 0.477 | 40.0 | 0.023 |
| Telemetry LR | 0.587 | 0.487 | 49.5 | 0.043 |
| Random | 0.500 | 0.327 | 51.2 | 0.072 |

sparsity. Progress features also carry temporal structure, activating later within episodes, which enables prefix-level monitoring and the dense auxiliary reward use case.

## APPENDIX U
### SPEED-HAZARD THRESHOLD SENSITIVITY

Section VI reports the high-approach-speed category at chance (0.535 AUROC), and re-thresholding does not rescue it. Table XXXI sweeps the hazard threshold $0.8\times$–$1.2\times$ the default (560 episodes per setting). The speed-hazard *positive rate* swings from 0.921 at $0.8\times$ to 0.318 at $1.0\times$ to 0.068 at $1.2\times$, with the lowest label consistency of any category. The proxy never defines a stable hazard event, so the at-chance result reflects an ill-posed proxy rather than a tuning artifact, and a logged real-world speed event should replace it before any monitoring claim.

## APPENDIX V
### REPRODUCIBILITY NOTES

All activations are cached from OPENVLA on LIBERO suites `goal`, `object`, `long`, and `spatial`. Differential feature tests use Mann–Whitney $U$ with Benjamini–Hochberg FDR control at $q = 0.05$, and readouts are $\ell_2$–regularized logistic regression. Safety evaluation uses leave-one-task-out splits and a 25-step future-violation window, with random-feature controls matched in count and activation sparsity to the ranked top features. The progress label is a suite-normalized geometric proxy, intentionally imperfect: a representation probe, not a final RL objective.

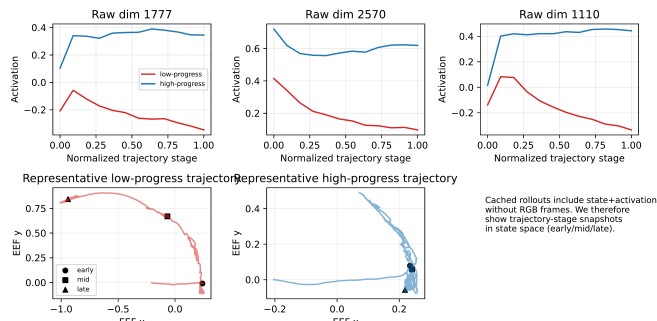

Fig. 8: Stage profiles of individual progress features over normalized episode time. High-progress features ramp up in later stages, while low-progress features dominate early, supporting prefix-level monitoring.

TABLE XXX: Top-ranked diagnostic features for the offline feature-setting sanity check.

| Rank | Feature | Active low | Active high | Mean low | Mean high | Higher |
|---|---|---|---|---|---|---|
| 1 | 10609 | 8.6% | 65.2% | 0.0013 | 0.0113 | high |
| 2 | 11471 | 7.5% | 66.8% | 0.0008 | 0.0093 | high |
| 3 | 1972 | 8.0% | 62.6% | 0.0008 | 0.0089 | high |
| 4 | 9215 | 9.1% | 66.3% | 0.0021 | 0.0144 | high |
| 5 | 11070 | 9.6% | 66.8% | 0.0016 | 0.0088 | high |
| 6 | 3154 | 65.8% | 3.2% | 0.0124 | 0.0002 | low |
| 7 | 11183 | 21.4% | 71.7% | 0.0015 | 0.0072 | high |
| 8 | 9998 | 67.4% | 4.3% | 0.0105 | 0.0003 | low |
| 9 | 15966 | 8.6% | 63.6% | 0.0005 | 0.0089 | high |
| 10 | 1528 | 9.1% | 62.0% | 0.0009 | 0.0061 | high |

TABLE XXXI: Hazard-label sensitivity to the detection threshold (560 episodes/setting). The high-approach-speed positive rate is the most threshold-sensitive and least consistent category, so its at-chance SAE AUROC persists under every threshold.

| Threshold | Speed pos. rate | Boundary pos. rate | Consistency vs. $1.0\times$ |
|---|---|---|---|
| $0.8\times$ | 0.921 | 0.998 | 0.558 |
| $0.9\times$ | 0.730 | 0.741 | 0.724 |
| $1.0\times$ | 0.318 | 0.289 | 1.000 |
| $1.1\times$ | 0.134 | 0.163 | 0.898 |
| $1.2\times$ | 0.068 | 0.107 | 0.848 |

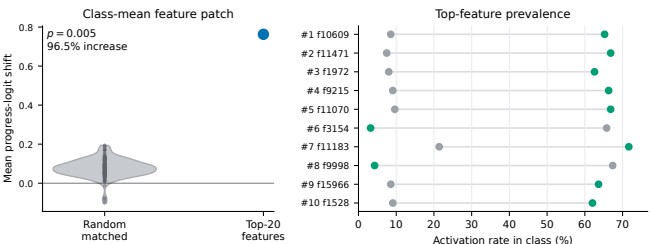

Fig. 9: Offline feature-setting sanity check. Setting top progress features to their high-progress means shifts the progress logit far more than matched random feature sets.