# OpenReview forum: "Sparse Progress and Risk Features in OpenVLA: Fragile Under Task-Family Shift"
_roboticsfoundation.org/RSS/2026/Workshop/RL4VLA — RL4VLA_

### Official Review · Reviewer_o8UW · 2026-06-26
**The paper introduces SAFESAE-VLA audits that evaluate the internal features of the VLA. The work introduces a new way to measure the internal state of a policy. But the relevance to the field of RL+VLA is not very clear, as the results don't back the usefulness for RL training/fine-tuning.**

**Rating:** 5
**Confidence:** 3

**Review:**

Sparse Progress and Risk Features in OpenVLA: Fragile Under Task-Family Shift
* The paper measures when internal representations break in VLAs and contributes a measurement and recipe to validate the internal representation. They thereby introduce SAFESAE-VLA audits that evaluate the internal features of the VLA before fine-tuning  by analyzing the states for decodable concepts, future violations, and intervention-relevant directions
* The authors run experiments with OpenVLA in Libero and show that the policy's internal features can decode task progress (high vs. low) with high accuracy in-distribution (0.932 AUROC)
* The paper does not introduce any new RL methods and just measures and analyzes the policy before fine-tuning or its internal state.
* In general, the authors are quite open about their results and show clearly their limitations and where their approaches fail, which helps to position their findings and increase the trustworthiness of the general results
* They claim that internal features survive task-level shift, but not family-level shift, and the RL induces a family shift. The first claim is reasonable and intuitive. The second claim is missing quantitative evidence that RL induces a family shift. That statement is not generally true. If we have a Replay Buffer that includes all family levels, we would not induce this RL shift. Of course, one would need to ensure that the Replay Buffer is optimized for family diversity and that the readout is re-ranked based on that coverage. A full-coverage task buffer addresses the family axis you measured, but not the behavioral-drift axis you also invoke, and you've quantified neither. To strengthen their main claim, the authors need to quantify it by showing that RL induces a family-level shift
* Also, the authors claim that RL for VLA is unusually fragile, which is not backed up by evidence or by literature
* The authors made the right observations that we need rewards that track the high-level progress instead of the motion. Therefore, they introduce the Progress Target, but their findings don't support the claim that this state helps learning with RL, even though they don't frame it as a perfect reward.
* The authors also introduce a prefix Safety Target to audit on hazard-proxy categories to check whether the policy can warn for potentially harmful states. But in their experiments, the authors only partially evaluate hazard-proxy detection and the AUROC score for those events, which are not very high, so there is no evidence that their approach can audit such scenarios. The detection claim is weak as it loses to a black-box baseline, though the veto as a rank-and-halt control does measurably reduce simulated violations
* The framing of some claims overreaches relative to the carefully-graded body, e.g., the safety and trainability language should be scoped down to match
* Overall, the paper is clearly written and clearly shows its limitations, but some statements are in contrast to its later limitations. The paper analyzes the inner behavior of VLA, which helps to shape a better understanding. In the special case of RL+VLA, the contribution is not very strong, as many experiments don't show the expected outcome, and some statements lack quantitative evidence. For example, the progress-weighted RL matched rather than beat the sparse-return baseline (0.650 vs 0.680) despite a near-ceiling 0.985 readout. The direction is interesting, but needs more analysis

---

### Official Review · Reviewer_xxm7 · 2026-06-27
**Sparse Progress and Risk Features in OpenVLA: Fragile Under Task-Family Shift**

**Rating:** 7
**Confidence:** 4

**Review:**

This paper applies BatchTopK sparse autoencoders (SAEs) to OpenVLA's residual-stream activations during closed-loop LIBERO manipulation to audit whether internal features encode progress and risk in ways useful for RL fine-tuning. The central positive finding is that progress is compactly decodable in-distribution, and progress/risk feature sets are nearly disjoint. The central—and most RL-relevant—negative finding is that this readout collapses to near-chance (0.537 AUROC) under family-held (cross-suite) distribution shift while surviving task-held (leave-one-task-out within families) shift at 0.933. The paper tests four downstream RL signals (curation, reward-shaping, live veto, offline RL weight update), reporting two as modestly positive and two as negative, with negative results foregrounded rather than buried. The work is rigorous and self-critical but limited to a single model, single simulator, and mostly offline/diagnostic evidence.

The paper is highly relevant to RL4VLA workshop on RL for vision-language-action models. The experimental quality is high and the statistical rigor is above average for this area. The offline RL experiment (Section V-F) is well-designed.

The paper is dense but well-organized. The abstract accurately previews both positive and negative results with key numbers. The contributions list uses a four-level grading vocabulary (diagnostic, operational, conjectured, refuted) carried through to Table XI's falsification criteria—a strong structural choice that makes the evidence-claim mapping transparent.

The application of SAEs to closed-loop VLA execution is novel. Prior SAE work targets language models almost exclusively.

This is a well-executed measurement paper with an important central finding (family-level collapse of progress readouts) and commendable scientific honesty. Its limitations—single model/simulator, mostly offline evidence, negative RL result, geometric proxy target—are real but openly acknowledged and bounded by explicit scope statements. The paper advances the field by providing a concrete audit recipe and by separating decodability from trainability. The practical RL impact is not yet demonstrated, but the diagnostic value is high and the negative results are as informative as the positive ones.

---

### Decision · Program_Chairs · 2026-07-03

**Decision:**

Accept

**Comment:**

This paper studies how OpenVLA represents task progress, showing that it can be captured by a compact sparse-feature basis but that performance drops to near chance under task-family shift. The reviewers appreciated the careful evaluation and the authors' willingness to highlight negative results instead of overselling the method, viewing the paper as a useful diagnostic contribution for RL4VLA. The main concerns are about clarity and presentation rather than the technical contribution. We believe these issues do not outweigh the overall contribution, and the paper is a valuable addition to the workshop. For the camera-ready version, the authors should improve the writing and fix unclear sentences, clearly state that the hazard labels are simulator-based proxies and not real-world safety measures, report the episode counts consistently in one table, and make the performance drop under task-family shift the main takeaway of the paper